# Expert Branches: Module Diversity for Stronger Feature Learning in Laparoscopic Segmentation

**Lin Guo**[*1]                    LIN.GUO@SLU.EDU
**Chiara Camerota**[*1]              CHIARA.CAMEROTA@SLU.EDU
**Mohammad Mahmoud**[2]            MMAHMOUD96@SIUMED.EDU
[2] *School of Medicine, Southern Illinois University, USA*
**Flavio Esposito**[1]              FLAVIO.ESPOSITO@SLU.EDU
[1] *Computer Science Department, Saint Louis University, USA*

**Editors:** Accepted for publication at MIDL 2026

## Abstract

Module diversity fundamentally enhances a model's ability to learn geometric structure by enabling a broader and more expressive set of feature representations. While many architectures improve performance by scaling parameters or relying on large-scale pretraining, these strategies make it difficult to identify which design principles truly enhance feature learning capability, especially in challenging domains with limited data such as laparoscopic surgical segmentation. This work investigates a parameter-constrained, no-pretraining setting to isolate the intrinsic feature learning capability of different module configurations. We introduce expert branches, a design concept that assigns different module families to their own independent pathways rather than mixing all features within a single stream. This separation encourages branch-specific specialization (Experts), reduces parameters, and avoids the entanglement that commonly obscures each module's contribution. We test this idea with TriEB, a UNet-based model incorporating CNN, deformable-convolution, and dynamic-snake branches with less total parameters. TriEB surpasses the vanilla UNet, the non-diverse TriCNN counterpart, and transformer-based models including SegFormer and Swin on the DSAD laparoscopic dataset. These results demonstrate that expert branches offer a more effective design principle for extracting diverse features from surgical imagery. The study highlights module diversity as a promising, architecture-agnostic framework for building efficient, interpretable, and data-adaptive feature extractors.

**Keywords:** Network design, Multi-Branch Network, Segmentation, Surgical Imaging.

## 1. Introduction

Feature extractors determine what geometric structure a vision model can perceive and how well it can generalize. This role becomes especially critical in laparoscopic image segmentation, where surgical scenes exhibit extreme visual complexity, including rapid camera motion, highly deformable organ surfaces, specular reflections, smoke, bleeding, tool occlusions, and thin, sharply curved anatomical structures. These conditions create severe appearance variability and weak global context, making segmentation fundamentally a problem of robust, geometry-aware feature learning under uncertainty. Yet no single feature extraction module is universally suited for such diverse geometric regimes: different architectural modules favor different structural patterns. Most existing architectures nevertheless collapse all

---

[*] Contributed equally

features into a single dominant pathway. This motivates a feature extraction strategy that embraces module diversity by design, allowing heterogeneous extractors to specialize rather than compete.

To address this limitation, we propose Expert Branches (EB), a general architectural abstraction for building module-diverse feature extractors. An Expert Branch is defined as an independently operating feature-extraction pathway that maintains its own computational stream and exposes its output only at explicit fusion points. By preventing intermediate tensor sharing across branches, EB encourages each branch to cultivate unique and complementary representations rather than converging toward redundant filters—a common outcome when many channels originate from the same entangled layer. Importantly, EB imposes no restriction on the internal structure of each branch: any module type, network block, or architectural family may serve as an expert. This makes EB a general-purpose design principle rather than a convolution-specific construction.

In this work, our goal is not to chase the highest possible segmentation accuracy through architectural scale or external knowledge, but to isolate and verify the intrinsic effect of module diversity on feature learning. We therefore adopt a highly constrained experimental setup: (i) minimal structural change to the backbone, (ii) compact parameter budget, and (iii) no external pretraining. Under this controlled setting, any observed performance gain can be directly attributed to how features are learned, rather than to model size or transferred representations.

We conducted the verification experiment on a simple realization, TriEB: a UNet backbone for laparoscopic organ segmentation. TriEB assigns three different convolutional module types to three independent expert branches, which process the same input but evolve separately and interact only through lightweight fusion modules. This design ensures that feature diversity arises explicitly from module diversity, rather than from widened channels within a single transformation family. Importantly, the use of three convolution types here serves strictly as a methodological probe to validate the EB principle, not as a claim that these particular operators form an optimal set. Specifically, standard convolutions tend to capture stable textures and low-frequency patterns, deformable convolutions (DCN) adjust sampling positions to accommodate nonrigid or spatially misaligned anatomy (Dai et al., 2017; Zhu et al., 2019), and dynamic snake convolutions (DSC) incorporate geometric priors that emphasize curved or boundary-sensitive structures (Qi et al., 2023).

Transformer-based vision models have advanced rapidly since ViT showed that pure attention architectures can rival CNNs under large-scale pretraining (Dosovitskiy et al., 2021b). Subsequent studies in medical imaging consistently report that their strong performance depends heavily on extensive pretraining or large in-domain datasets (Shamshad et al., 2023; Takahashi and colleagues, 2024). In practice, however, surgical datasets are often small and highly domain specific, limiting the ability of ViT-style models to learn rich low-level geometric features from scratch. Although segmentation architectures such as SegFormer and Swin (Xie et al., 2021; Liu et al., 2021) achieve impressive results with strong pretraining, their global attention mechanisms do not inherently guarantee expressive boundary- or deformation-aware features under data-limited conditions, consistent with recent findings in medical self-supervised learning (Huang et al., 2023; Zeng and colleagues, 2024). Accordingly, all transformer baselines in this work are trained from scratch on

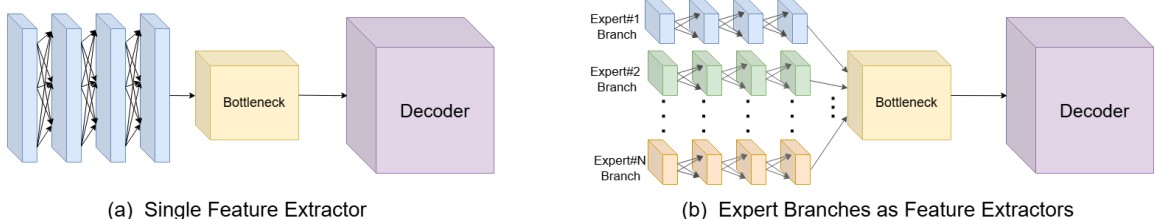

(a) Single Feature Extractor  (b) Expert Branches as Feature Extractors

Figure 1: Structural comparison between (a) a conventional single-stream feature extractor that fully entangles all channels within one shared pathway, and (b) our Expert Branch architecture, which separates heterogeneous feature extractors into independent pathways.

DSAD (Carstens et al., 2023) to enable a fair comparison of intrinsic feature learning capacity. Pretrained transformer experts will be explored in future EB systems.

When evaluated on the challenging DSAD laparoscopic dataset, TriEB consistently outperforms the vanilla UNet (Ronneberger et al., 2015), baseline multi-branch variants built from a single operator family, and transformer-based baselines trained from scratch, demonstrating a stronger ability to capture complex geometric cues directly from limited-domain data. The independent branches further exhibit interpretable specialization behavior, revealing how different experts contribute to distinct anatomical structures. Expert Branches provide a simple yet powerful abstraction for building module-diverse, data-adaptive feature extractors by explicitly separating heterogeneous feature learners into independent computational routes.

Unlike a fully developed network targeting state-of-the-art performance, TriEB serves as a minimal and deliberately constrained validation of the Expert Branch principle, designed to isolate the intrinsic impact of module diversity on feature learning and to lay the foundation for future EB frameworks that integrate broader architectural families, including pretrained transformer experts, for more advanced surgical scene understanding.

## 2. Related Work

Multi-branch architectures have been widely explored to enhance representational diversity, including Inception networks (Szegedy et al., 2015), frequency-decomposed hybrids, and convolution–attention mixtures such as ACmix (Pan et al., 2023). While these designs expand feature diversity, features from different operators are typically merged early, producing entangled representations that limit interpretability. Moreover, because branches interact through shared channel flows, pruning and scaling often require disruptive architectural changes.

In contrast, our work organizes heterogeneous operators into independent expert branches. By preserving the identity of each operator throughout the encoder rather than mixing features immediately, this design enables specialization, operator-aware analysis, and predictable, structurally safe pruning.

Laparoscopic and endoscopic images pose challenges distinct from those in conventional medical imaging. Real surgical scenes involve rapid camera motion, instrument occlusions, specular highlights, smoke, bleeding, and extreme nonrigid deformations. Anatomical struc-

tures often appear only partially and change appearance as instruments interact with tissue, while thin, tortuous boundaries such as vessels and ducts are particularly difficult to recover with a single operator family.

Most surgical segmentation pipelines are built upon convolutional networks, including UNet variants (Ronneberger et al., 2015; Çiçek et al., 2016). Standard convolutions provide efficient local feature extraction but are limited in modeling geometric variability. Deformable convolutions (DCN) (Dai et al., 2017; Zhu et al., 2019) improve robustness to shape changes through adaptive sampling, and Dynamic Snake Convolution (DSC) (Qi et al., 2023) embeds geometric priors to enhance curved and boundary-focused structures. These operators exhibit complementary strengths, yet existing systems usually adopt only one operator family or combine them within shared blocks where representations quickly become entangled, obscuring operator contributions and complicating structured pruning.

Vision transformers (ViTs) have emerged as a dominant paradigm in computer vision (Dosovitskiy et al., 2021a; Liu et al., 2021; Raghu et al., 2021; Li et al., 2024), but their strong performance relies heavily on large-scale pretraining. In medical and surgical imaging, where datasets are smaller and visually distinct, ViT models trained from scratch often underperform strong convolutional baselines, particularly on thin structures and irregular boundaries (Xie et al., 2021; Liu et al., 2021; Chen et al., 2023). This highlights a fundamental limitation: without extensive pretraining, transformers struggle to learn domain-specific geometric cues directly from limited data.

These limitations share a common root: effective surgical segmentation requires feature extractors that can jointly model stable regional appearance, large nonrigid deformation, and fine curvilinear boundaries. Yet most existing architectures either commit to a single dominant module type or entangle heterogeneous features too tightly to preserve their individual strengths. The Expert Branch strategy pursued in this work directly addresses

## 3. Methodology

### 3.1. Expert Branches: A Concept of Module Diversity

The core idea of this work is the principle of Expert Branches (EB), which organizes heterogeneous feature extraction modules into disentangled and independent pathways rather than mixing all feature channels within a unified shared stream. Different modules exhibit distinct inductive biases and are naturally suited to different geometric regimes. By separating them into dedicated branches, the model allows each module to express its representational strengths without interference from others. As a result, EB maintains a diverse and complementary set of geometric representations, rather than forcing all features to conform to a single transformation family.

The Expert Branch concept departs from conventional architectures that increase capacity by widening layers or deepening a single dominant module type. Such designs force all channels to propagate through the same feature transformation, even though many visual cues may not be well modeled by that module. Moreover, blending heterogeneous modules within the same block—such as through naive concatenation or tightly coupled attention-based fusion—entangles their effects, obscures their individual functional roles, and makes structural pruning unpredictable. Expert Branches instead establish clean, module-specific

Table 1: Approximate parameter breakdown (in millions). Totals match the overall model sizes: Vanilla UNet ≈ 30.76M, TriEB_CC ≈ 24.98M, TriEB_WF ≈ 18.8M.

| Component | Vanilla UNet | TriEB-CC | TriEB-WF |
|---|---|---|---|
| Stem | 0.04 | 0.03 | 0.01 |
| Encoder Stage 1 | 0.22 | 0.19 | 0.19 |
| Encoder Stage 2 | 0.88 | 0.70 | 0.70 |
| Encoder Stage 3 | 3.54 | 2.73 | 2.73 |
| Encoder Stage 4 | 14.16 | 10.77 | 10.77 |
| Fusion modules | 0.00 | 0.00 | 1.4 |
| Decoder (all stages) | 11.92 | 10.57 | 2.98 |
| **Total** | **30.76** | **24.98** | **18.8** |

feature streams with explicit fusion points, enabling the network to learn how different experts should collaborate without artificially coupling their internal representations.

Parameter efficiency is another key motivation of the EB design. Because each branch is narrower than a full-width backbone encoder, the model avoids the quadratic parameter growth associated with uniformly widening convolutional layers. Even with multiple expert branches, the total parameter count remains comparable to or lower than that of the original single-stream backbone. At the same time, the diversity of modules creates a richer geometric basis for feature learning, allowing the network to capture structures that a single module family could not represent efficiently. This balance between reduced per-branch capacity and enhanced representational diversity underlies the practical strength of the Expert Branch framework.

### 3.2. TriEB: Minimal Expert Branch Instantiation, Bottleneck Behavior, and Fusion

We implement the Expert Branch concept within a UNet encoder to form TriEB, a minimal three-branch realization designed specifically for controlled methodological verification. UNet is selected as the backbone due to its structural simplicity, extensive prior study, and its ability to be systematically scaled down to meet strict parameter constraints. This property is critical for the present study, as increasing the overall parameter budget would confound the analysis by making it unclear whether observed performance gains arise from module diversity or from increased model capacity. In this setup, three different convolutional module types are assigned to three independent expert branches. This configuration is not intended to define an optimal operator combination, but rather to serve as a compact and interpretable experimental instantiation that introduces feature diversity with minimal structural change to the backbone. The underlying premise is that features learned within a single transformation family are inherently limited, whereas diversity across feature extraction sources can yield more expressive and complementary representations. At each resolution level, all branches receive the same input feature map and process it independently without sharing intermediate activations. Interaction occurs only through lightweight

fusion modules, preserving branch-level specialization while maintaining a compact encoder footprint.

In this initial study, we deliberately restrict TriEB to convolutional modules as a controlled testbed for validating the Expert Branch principle. Standard convolutions (CNN), deformable convolutions (DCN), and dynamic snake convolutions (DSC) represent three distinct operator families with complementary inductive biases, and all have been shown to function effectively within the same UNet-style encoder. This makes them suitable candidates for studying whether potentially supplementary experts can collectively enhance feature diversity under a unified backbone. Vision transformers are excluded as expert branches in this study not only because their architectural form differs fundamentally from convolutions and would substantially increase the overall parameter budget, but also because their strong performance is known to rely heavily on large-scale pretraining. Since the objective of this work is to evaluate features that are learned intrinsically under constrained and data-limited conditions, all models are trained from scratch to isolate feature learning capacity without external knowledge transfer. Incorporating pretrained transformer modules as additional expert branches is therefore left as a natural direction for future work.

Within this constrained setting, the three convolutional branches serve as one experimental realization for generating feature diversity through heterogeneous inductive biases. One branch uses standard convolutions (CNN) to model stable regional appearance, a second branch employs deformable convolutions (DCN) to adapt to nonrigid or spatially misaligned structures, and a third branch applies dynamic snake convolutions (DSC) to emphasize curved or boundary-sensitive patterns. To further isolate the effect of expert diversity, we conduct ablation studies using TriCNN, TriDCN, and TriDSC, where only a single operator type is used across all three branches under the same TriEB-WF fusion structure. The vanilla UNet baseline uses a base channel width of 64, while all TriEB-based models use a base width of 32.

Separating feature extractors into dedicated expert branches reshapes both the representational behavior and the capacity distribution of the encoder. Each branch is intentionally narrow: instead of allocating full-width channels to a single stream, TriEB redistributes capacity into multiple compact branches. Although each branch is narrower in isolation, their joint representation becomes more expressive. Before fusion, the concatenated feature tensor spans 96 channels, forming a broader intermediate bottleneck that preserves module diversity. This structural redistribution also leads to improved parameter efficiency. As shown in Table 1, TriEB reduces the parameter count from approximately 31M (vanilla UNet) to 25M in the channel-concatenation variant (TriEB-CC) and 19M in the weighted-fusion variant (TriEB-WF), while maintaining strong internal expressiveness. These properties also make Expert Branches naturally compatible with structured pruning and efficient scaling.

TriEB supports two fusion mechanisms at each encoder stage. The first, denoted TriEB-CC (channel concatenation), concatenates the three branch outputs,

$$\mathbf{Y}_{\text{CC}}^{(s)} = [\,\mathbf{F}_{\text{cnn}}^{(s)} \,\|\, \mathbf{F}_{\text{dcn}}^{(s)} \,\|\, \mathbf{F}_{\text{dsc}}^{(s)}\,],$$

followed by a $1 \times 1$ convolution for channel compression. This strategy preserves branch independence, maintains the largest bottleneck space, and is the most stable under pruning because feature streams remain disentangled. The second mechanism, TriEB-WF (weighted

fusion), uses learned weights to combine the three branches,

$$\mathbf{Y}_{\text{WF}}^{(s)} = \sum_{t \in \{c,d,s\}} g_t^{(s)} \mathbf{F}_t^{(s)}, \tag{1}$$

$$g^{(s)} = \text{softmax}\left(W^\top \text{GAP}\left([\mathbf{F}_t^{(s)}]\right)\right). \tag{2}$$

This fusion is more compact and parameter-efficient but introduces inter-branch dependencies, making structural pruning more sensitive.

Formally, at encoder stage $s$ with input feature map $\mathbf{X}^{(s)}$, the three expert branches compute

$$\mathbf{F}_{\text{cnn}}^{(s)} = f_{\text{CNN}}^{(s)}(\mathbf{X}^{(s)}), \tag{3}$$

$$\mathbf{F}_{\text{dcn}}^{(s)} = f_{\text{DCN}}^{(s)}(\mathbf{X}^{(s)}), \tag{4}$$

$$\mathbf{F}_{\text{dsc}}^{(s)} = f_{\text{DSC}}^{(s)}(\mathbf{X}^{(s)}). \tag{5}$$

The fused output is then passed to the decoder, allowing complementary representations from heterogeneous experts to be combined adaptively for downstream segmentation.

## 3.3. Pruning as Expert Allocation

Rather than treating pruning as a post-hoc compression step, we integrate it into the EB framework as an *expert allocation* mechanism. All expert branches are initialized at full width and trained with mild sparsity regularization to encourage suppression of redundant channels. After convergence, channel saliency is estimated using the L1 norm of filter weights, followed by a single unstructured pruning pass that removes the lowest-scoring channels within each branch. We used 30% pruning ratio as a common aggressive-but-stable sparsity target.Because Expert Branches are architecturally independent and do not share intermediate activations, pruning does not introduce shape mismatches and can safely suppress weak channels or even entire branches. A brief fine-tuning stage of 25 epochs with a reduced learning rate is then applied to recover any accuracy loss. This process transforms pruning from a deployment optimization into a tool for analyzing the relative contributions of different experts.

## 3.4. Metrics

We report both region-based and boundary-based metrics to comprehensively evaluate segmentation performance.

**Dice and mIoU (region-based).** Dice and mean Intersection-over-Union (mIoU) quantify the pixel-wise overlap between predicted and ground truth masks and are widely used to assess volumetric segmentation accuracy. These metrics primarily reflect how well the model captures the overall extent of each anatomical region.

**MASD (Mean Average Surface Distance).** MASD measures the average geometric distance between the predicted and ground truth organ boundaries in both directions. It directly evaluates how far the predicted surface deviates from the true anatomical contour and provides a physically interpretable measure of boundary accuracy in pixel units (Maier-Hein et al., 2024).

**NSD (Normalized Surface Dice).** NSD measures the fraction of boundary points that lie within a predefined tolerance of the ground truth surface, normalized by the total boundary length (Maier-Hein et al., 2024). It reflects the proportion of the organ contour that is correctly localized within a clinically acceptable margin. In laparoscopic surgery, organ

boundaries guide critical tasks such as dissection, exposure, and the preservation of delicate structures. While region-based metrics such as Dice and mIoU quantify volumetric correctness, they do not strongly penalize thin boundary deviations. In contrast, MASD and NSD directly evaluate how well a model captures fine anatomical contours—an essential requirement because organ boundaries are often thin, curved, and partially occluded; surgical instruments interact with tissues precisely along these boundaries; even small pixel-level errors can indicate incorrect delineation of vital structures; and many organs exhibit similar textures but distinct shape-driven signatures. Therefore, boundary-based metrics provide a more clinically meaningful assessment of segmentation quality in surgical scenes, where accurate delineation is crucial for downstream decision support and intraoperative guidance.

## 4. Results

### 4.1. Experimental setup and training details.

In all experiments, we use a vanilla UNet encoder with four resolution stages and a base channel width of 64, and a TriEB encoder with the same number of stages but a reduced base width of 32. This configuration ensures that the total encoder parameter count of TriEB is comparable to, but slightly smaller than, that of the UNet baseline, allowing observed performance gains to be attributed to module diversity rather than increased model capacity, as shown in Tab. 1. All models are optimized using Adam and are trained for a fixed schedule of 80 epochs, by which convergence is consistently observed. The UNet and TriEB models are trained with a learning rate of $1 \times 10^{-4}$ and a weight decay of $1 \times 10^{-3}$, the transformer-based models(SegFormer and Swin) are trained a learning rate of $6 \times 10^{-5}$ and a weight decay of $1 \times 10^{-2}$. Although DSAD provides a validation split, early stopping is not employed to ensure that all models are trained for an identical number of epochs under the same conditions, enabling fair comparison. Since all networks share the same encoder–decoder structure and number of layers, inference time remains at the millisecond level for all methods, and differences between models are negligible.

### 4.2. Overall Performance

Table 2 reports the average segmentation performance across all 11 DSAD organs, with qualitative examples shown in Fig. 2. Since all models are trained from scratch, the results directly reflect the intrinsic feature learning capability of each architecture without reliance on large-scale pretraining.

Both Expert Branch designs, TriEB-CC and TriEB-WF, outperform the vanilla UNet baseline and the transformer-based models (SegFormer and Swin) trained from scratch across nearly all metrics. The improvements are most consistent in mIoU, mDice, and NSD, demonstrating that explicit module diversity leads to stronger region-level accuracy

Table 2: Overall laparoscopic segmentation performance on DSAD averaged across 11 organs. Higher mIoU, mDice, and NSD and lower MASD indicate better performance. Ablation models (TriCNN, TriDCN, TriDSC) use a single convolution operator replicated across all branches to isolate the effect of module diversity. UNet serves as the single-stream baseline. Pruned models apply 30% unstructured pruning & fine-tune after convergence. Transformer baselines (SegFormer and Swin) are trained from scratch. An asterisk (*) indicates statistical significance compared to the UNet baseline.

| Model | Setting | mIoU | mDice | MASD↓ | NSD↑ |
|---|---|---|---|---|---|
| TriEB-WF | Pruned | 0.456 | 0.542 | 27.133 | 0.449 |
| TriEB-WF | Original | 0.454 | 0.541 | 20.439 | 0.432 |
| TriEB-CC | Pruned | 0.471 | 0.559 | 25.442 | 0.467* |
| TriEB-CC | Original | 0.471 | 0.564 | 17.881* | 0.450* |
| TriCNN (ablation) | Original | 0.410 | 0.497 | 24.116 | 0.391 |
| TriDCN (ablation) | Original | 0.421 | 0.511 | 20.747 | 0.405 |
| TriDSC (ablation) | Original | 0.436 | 0.528 | 22.084 | 0.415 |
| UNet (baseline) | Pruned | 0.438 | 0.530 | 28.000 | 0.411 |
| UNet (baseline) | Original | 0.445 | 0.537 | 21.337 | 0.411 |
| SegFormer (scratch) | Original | 0.357 | 0.446 | 26.539 | 0.298 |
| Swin (scratch) | Original | 0.382 | 0.476 | 23.289 | 0.326 |

and more reliable boundary localization. Compared to the single-operator ablation models (TriCNN, TriDCN, and TriDSC), the full TriEB designs achieve higher overall performance, confirming that the gains arise from heterogeneous expert collaboration rather than from any individual operator alone. Statistical significance is evaluated across all 11 organs, and while improvements are not uniformly observed for every organ, this behavior is expected: in cases where standard convolutions (as used in the UNet baseline) are already sufficient to capture the underlying structure (as shown in Fig. 2 Liver), performance gains are naturally limited. TriEB provides its greatest benefit in scenarios where single-operator CNN features struggle, leveraging complementary expert representations to handle more challenging anatomical conditions, an effect that is particularly evident in boundary-focused metrics.

Boundary-focused metrics further highlight the advantage of Expert Branches. Both TriEB variants achieve the highest NSD among all models, indicating more precise anatomical contour alignment. This supports the motivation behind EB: independent experts capture complementary geometric cues that improve the representation of thin, curved, and deformable structures that dominate laparoscopic scenes. In contrast, transformer baselines trained from scratch show substantially weaker performance across both region-based and boundary-based metrics, underscoring the difficulty of learning fine-grained geometric structure from limited surgical data without pretraining.

Pruning reveals additional insights into architectural robustness. For both TriEB variants, pruning has minimal impact on mIoU and mDice and slightly improves NSD, indicating that boundary localization remains stable under moderate sparsification. In contrast,

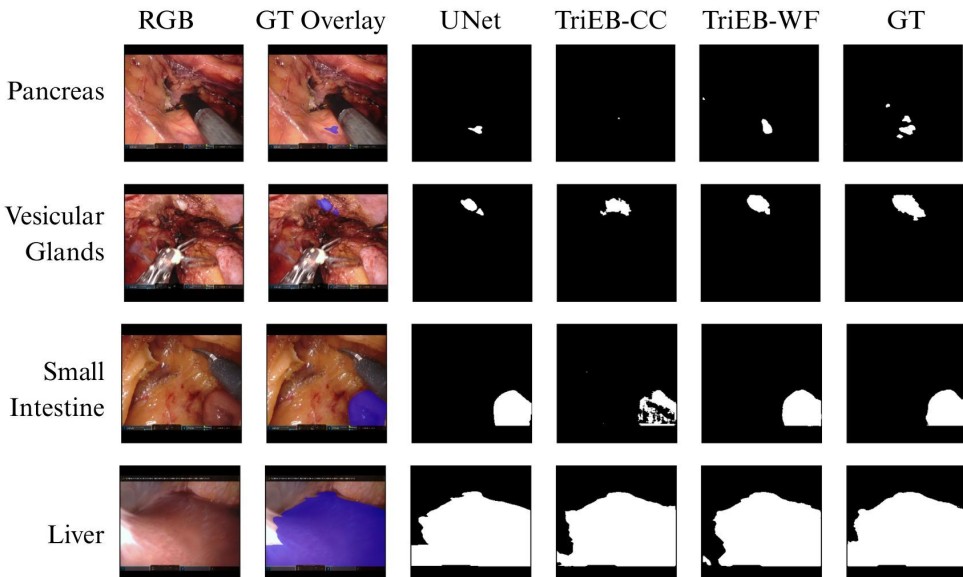

Figure 2: Qualitative comparison of segmentation results on representative DSAD images. UNet performs well on visually prominent and structurally simple organs (e.g., liver, vesicular glands) but shows reduced accuracy on smaller structures (e.g., pancreas) and organs with weak color cues (e.g., small intestine). TriEB models better preserve anatomical structure and boundary continuity in these challenging cases.

the UNet baseline exhibits larger and less predictable changes after pruning, consistent with its fully entangled channel structure. Across models, MASD tends to increase after pruning, reflecting the emergence of occasional long-range false positives that strongly affect surface distance while minimally impacting region-based scores. This mixed behavior indicates that pruning sharpens true boundary representation while introducing isolated surface outliers.

### 4.3. Organ-Wise Evaluation

Table 3 reports organ-wise mIoU for all unpruned models trained from scratch. TriEB-CC achieves the highest accuracy in the majority of organs, with particularly strong improvements in anatomically complex structures such as the colon, stomach, liver, and small intestine. These organs exhibit steep curvature, variable shape, and strong nonrigid deformation—precisely the regimes where module diversity is expected to provide an advantage.

TriEB-WF performs competitively across most organs, indicating that even without maximizing the intermediate bottleneck space through full channel concatenation, allocating heterogeneous modules to independent branches remains effective. Both Expert Branch variants consistently outperform SegFormer and Swin when trained from scratch, reinforcing the observation that transformer tokenization alone does not guarantee strong low-level geometric feature learning under limited data.

Several organ-level trends are evident. Expert Branches yield broad improvements across organs with diverse geometric characteristics, ranging from smooth surfaces (e.g., abdominal wall) to highly deformable and shape-complex regions (e.g., colon and stomach). Pruning

Table 3: Organ-wise mIoU for unpruned models on DSAD. All models are trained from scratch.

| Organ | UNet | TriEB-WF | TriEB-CC | SegFormer | Swin |
|---|---|---|---|---|---|
| Vesicular glands | 0.217 | 0.215 | 0.216 | 0.114 | 0.122 |
| Spleen | 0.586 | 0.661 | 0.649 | 0.480 | 0.532 |
| Colon | 0.534 | 0.568 | 0.587 | 0.471 | 0.523 |
| Small intestine | 0.709 | 0.719 | 0.746 | 0.644 | 0.646 |
| Stomach | 0.484 | 0.542 | 0.561 | 0.392 | 0.415 |
| Pancreas | 0.187 | 0.192 | 0.210 | 0.141 | 0.162 |
| Abdominal wall | 0.758 | 0.760 | 0.769 | 0.712 | 0.708 |
| Intestinal veins | 0.361 | 0.315 | 0.328 | 0.279 | 0.283 |
| Liver | 0.516 | 0.603 | 0.608 | 0.430 | 0.492 |
| Inf. mes. art. | 0.266 | 0.269 | 0.296 | 0.188 | 0.211 |
| Ureter | 0.276 | 0.145 | 0.213 | 0.0810 | 0.105 |

analysis at the deepest encoder stage (Stage 4) shows that standard CNN features are consistently preserved across organs, while DCN and DSC branches exhibit higher and more variable sparsity, indicating adaptive supportive roles. Across all 11 organs at Down4, the mean (variance) pruning sparsities for CNN, DCN, and DSC are 39.3% (0.08%), 84.1% (6.26%), and 46.8% (4.15%), respectively, underscoring their distinct and complementary roles within the encoder. The detailed table is presented in the Appendix. This organ-dependent variation in sparsity suggests that expert branches collaborate through complementary representations rather than redundancy, leading to stronger cross-organ robustness and improved geometric generalization. While segmentation of extremely small and thin structures such as the ureter remains challenging, Expert Branches still achieve competitive or improved performance relative to transformer baselines trained from scratch. Overall, these results confirm that Expert Branches enhance intrinsic feature learning and geometric representation without relying on increased model scale or external pretraining.

## 5. Conclusion

This work is grounded in the premise that feature learning from a single module family is intrinsically constrained, as each operator encodes a specific and incomplete set of inductive biases. Expert Branches address this limitation by introducing explicit module diversity, enabling the network to learn complementary feature representations under controlled conditions rather than relying on scale or pretraining. In this study, we validated the Expert Branch principle through TriEB, a deliberately constrained UNet-based realization designed to isolate the intrinsic impact of module diversity on feature learning in laparoscopic segmentation. Under a fixed parameter budget and without external pretraining, TriEB consistently outperformed a single-stream UNet, single-operator multi-branch variants, and transformer-based models trained from scratch, particularly on geometrically complex and boundary-sensitive structures. These results demonstrate that disentangling heterogeneous feature extractors into independent branches provides a principled and effective means of enhancing intrinsic feature learning capability, even when individual branches are compact.

While the present work focuses on a UNet-based realization with convolutional expert branches as a controlled testbed, the Expert Branch framework itself is not restricted to a specific backbone, operator family, or number of branches. Because each expert branch operates independently and interacts with others only through explicit fusion points, EB can be naturally adapted to a wide range of encoder architectures, including alternative convolutional backbones (e.g., DeepLab-style encoders, encoder–decoder hybrids, or hierarchical CNNs) as well as transformer-based or hybrid designs. Moreover, the number and composition of expert branches can be flexibly adjusted according to task requirements, allowing EB to scale from a small set of complementary modules to richer expert collections. In future work, we will explore integrating non-convolutional experts, including vision transformer branches, where each expert can be pretrained independently to maximize its representational strength prior to integration, enabling EB to support more general, scalable, and architecture-agnostic surgical scene understanding.

## Acknowledgments

This work has been supported by SIU School of Medicine and by NSF award OAC 2430236 and OAC 2201536.

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

## Appendix A. Organ-wise Pruning Analysis at Deep Encoder Stages

This appendix reports detailed organ-wise pruning sparsity statistics for the CNN, DCN, and DSC expert branches at the deepest encoder stage (Encoder Stage 4). While standard convolutional features exhibit relatively stable sparsity across organs, deformable and dynamic snake convolution branches show substantially higher and more variable pruning levels, indicating adaptive utilization across different anatomical structures. These organ-dependent pruning patterns provide further evidence that Expert Branches collaborate through complementary contributions. The table in this appendix supports the interpretability analysis discussed in the main text.

Table A1: Organ-wise pruning sparsity (%) at encoder stage Encoder Stage 4 across expert branches. Higher values indicate more aggressive pruning.

| Organ | CNN | DCN | DSC |
|---|---|---|---|
| Vesicular glands | 39.55 | 84.20 | 44.10 |
| Spleen | 39.25 | 85.40 | 49.60 |
| Colon | 39.00 | 87.95 | 47.95 |
| Small intestine | 39.55 | 84.20 | 44.10 |
| Stomach | 39.15 | 87.15 | 46.97 |
| Pancreas | 39.10 | 82.40 | 47.42 |
| Abdominal wall | 39.80 | 81.10 | 43.32 |
| Intestinal veins | 39.70 | 81.25 | 48.42 |
| Liver | 39.15 | 87.15 | 46.97 |
| Inf. mes. art. | 39.00 | 87.95 | 47.95 |
| Ureter | 39.10 | 82.40 | 47.42 |
| Mean | 39.3 | 84.1 | 46.8 |
| Variance | 0.08 | 6.26 | 4.15 |

