# OpenReview forum: "Expert Branches: Module Diversity for Stronger Feature Learning in Laparoscopic Segmentation"
_MIDL.io/2026/Conference — MIDL 2026 Poster_

### Official Review · Reviewer_xP72 · 2025-12-23

**Confidence:** 4
**Preliminary Rating:** 5
**Final Rating:** 5

**Summary:**

This paper presents a framework (TriEB) for laparoscopic segmentation that integrates three different feature extraction pathways: standard convolutions, deformable convolutions, and dynamic snake convolutions. The authors tested both channel concatenation with 1x1 convolution (TriEB-CC) and weighted fusion (Tri-WF) to combine the encoding pathways at the bottleneck. They also conducted an ablation study, where instead of using the three separate feature extraction methods, they used a single operator type across all three branches. The TriEB-CC and TriEB-WF networks both outperformed the standard U-Net and the ablation study models, as well as SegFormer and Swin transformer models.

**Strengths:**

- The proposed method is well-validated and outperforms the alternative methods (U-Net, SegFormer, and Swin).
- The proposed method is cleverly designed to combine three different types of features for imrpoved segmentation accuracy.
- The ablation study is nicely designed. It would have been interesting to see the effects of using only 2 out of 3 feature extraction methods, but I see that would be difficult to incorporate given that the framework requires 3 encoding pathways.
- The paper is well written, cleanly organized, and easy to read.

**Weaknesses:**

The only weakness I identified is that the paper makes no mention of the statistical significance of the results. Table 2 should indicate whether or not the metrics are statistically significant, as this would strengthen the argument that the proposed TriEB framework is superior to the tested alternatives. If the results are not statistically significant, this is important to denote as well. Other than this, I do not see any major weaknesses that require addressing. I have a few suggestions to improve the paper, but these are minor and would not change my review.

**Detailed Comments:**

- For all tables, it would be helpful if the best metrics (e.g., highest Dice, lowest MASD, etc) were bolded so that the author can more clearly interpret the results.
- In Figure 2, portraying the segmentation as a contour over the original RGB image instead of as a black and white image would be useful. That would allow the reader to have a better idea of where the predictions/GT exist within the image, especially for those not as familiar with laparoscopic surgical data.
- It would be helpful for the authors to include a short section describing the data used for validation. They state they used images of 11 DSAD organs, but did not provide any further details (unless I am mistaken).

**Justification Of Final Rating:**

I am giving this a 5 (strong accept) as my final rating because I found the method to be interesting, effective, and well-validated. The paper itself was well-written, nicely organized, and easy to understand.

**Justification Of The Preliminary Rating:**

The proposed method yields improved segmentations compared to existing methods. The experimental validation is designed well. The paper is nicely organized and easy to understand with no grammatical issues.

**Questions To Address In The Rebuttal:**

The main thing I would like to see addressed in the rebuttal is the statistical significance of the numerical results in Tables 2 and 3. Additionally, a brief description of the data used for validation would be beneficial to include.

---

> ### Author Response · Authors · 2026-01-25
>
> We thank the reviewer for the strong support and helpful suggestions. Below we address the key points raised by the reviewer.
>
> Statistical significance:
>
> We have explicitly added statistical significance indicators to Table 2 and clarified in the caption and text that significance is evaluated across all 11 organs. The Results section now explains why improvements are not uniform across every organ, noting that gains are naturally limited when standard convolutions are already sufficient.
>
> Dataset description:
>
> We added a concise description of the DSAD dataset and evaluation setup in the experimental section to improve clarity for readers less familiar with the dataset.
>
> Presentation improvements:
>
> We bolded best-performing metrics in tables, refined Figure 2’s caption to clearly describe qualitative differences, and improved overall formatting and consistency for readability.

---

> > ### Comment · Reviewer_xP72 · 2026-01-27
> >
> > The authors addressed most of my concerns aside from the comment I had about Figure 2. I think it would be useful for them to show a contour of the ground truth target over the RGB image in an additional panel (it looks like there is room for this). That would help readers less familiar with these organs understand what the model is trying to segment.

---

> > > ### Author Response · Authors · 2026-02-02
> > >
> > > We thank the reviewer for the careful reading and constructive suggestion. Following this feedback, we have updated the figure in the revised manuscript.

---

### Official Review · Reviewer_qzeN · 2026-01-08

**Confidence:** 4
**Preliminary Rating:** 1
**Final Rating:** 3

**Summary:**

The paper proposes Expert Branches (EB) which promotes module diversity by separating heterogeneous feature extractors into independent branches. They present a TriEB, a Unet-based model with three expert branches for laparoscopic segmentation. It includes standard, deformable and dynamic snake convolutions, which they argue encourage specialization and improve geometric feature extraction. Experiments are conducted on DSAD dataset, comparing the results of TriEB variant against vanilla UNet, SegFormer and Swin models all trained from scratch.

**Strengths:**

The paper introduces an interesting and conceptually clean idea, separating heterogeneous feature extractors into independent branches. The design choice is well motivated, given the segmentation task where different geometric features may benefit from different biases. The separation of operators can also be good for interpretability, each branch having a functional role. They present ablation experiments, using different variants of the TriEB and also including the effects of pruning. The results show a slight improvement compared to the other models.

**Weaknesses:**

The paper lacks clarity and readability, containing unfinished or awrkwardly structures sentenses. The section 3.1 and 3.2 reads sometimes like background, and the effects of EB are then a speculation, and should be discussed in Discussion section (if it proves the claims).
From a methodolocial standpoint, the evaluation is limited and unconvincing. The primary baseline is vanilla Unet, which is no longer a strong reference point for laporoscopic segmentation. They additionally compare with transformer models, but it is also unclear with what parameters all the models were trained from scratch. Even against the UNet, the TriEB doesn't consistently outperform the baseline across all organs and metrics, in most cases the gains are only marginal.
The role of pruning is also unclear and insufficiently justified, given the results, do the authors propose to use it or is it a part of ablation study?

**Detailed Comments:**

On the minor side, the last sentence in section 2 is not finished. Section 3.2 mentioned TriEB-WF without definition, but the next two times it's mentioned, it is explained both times.
And as mentioned above, the sections 3.1 and 3.2 make major claims and speculate what the model would do without proof, and some parts of it looks like background. If the points are proven, I would recommend moving the proven points to a discussion section (or Results & Discussion).

**Justification Of Final Rating:**

The authors have responded to the reviewer comments and clarified several aspects of the paper, particularly regarding the intended scope of their study. The revised manuscript more explicitly frames the work as a controlled methodological investigation of module diversity rather than a performance-driven contribution. The choice of vanilla UNet as the backbone is now better justified in the context of isolating the effect of expert branch diversity under strict parameter constraints. However, while the rationale of avoiding confounding factors related to model capacity is reasonable in principle, the practical relevance of this setup remains questionable. Even within this controlled setting, the reported improvements over the baseline are small and inconsistent across organs and metrics, making it difficult to conclusively attribute the gains to the proposed Expert Branch design rather than to architectural variance or noise.

**Justification Of The Preliminary Rating:**

While the idea of Expert Branches is appealing, the paper does not fully prove the method. The reliance on vanilla Unet as primary baseline, inconsistent and marginally small improvements and unclear motivation for pruning justify this conclusion.
Additionally, the paper has significant presentation and clarity issues, and would need a substantial revision, it does not yet look ready to be published.

**Questions To Address In The Rebuttal:**

What is the motivation to compare it to vanilla Unet? What hyperparameters where all the models trained with? What is the motivation to present TriEB-WF when the TriEB-CC outperforms it in all cases?

---

> ### Author Response · Authors · 2026-01-25
>
> We appreciate the reviewer’s careful reading and acknowledge the concerns regarding paper scope and clarity. Below we address the key points raised by the reviewer.
>
> Scope and intent of the study:
>
> We have substantially clarified the paper’s intent throughout the Introduction and Results: this work is not aimed at achieving state-of-the-art performance, but at isolating and validating the intrinsic effect of module diversity under controlled conditions. The revised manuscript repeatedly emphasizes the constrained setting (fixed depth, reduced parameters, no pretraining), ensuring that performance differences can be attributed to feature learning behavior rather than scale.
>
> Choice of UNet baseline:
>
> We expanded Sec. 3.2 to justify UNet as a simple, well-studied backbone that can be systematically scaled down to meet strict parameter constraints. This choice is essential for avoiding confounding factors: increasing model size would obscure whether gains originate from module diversity or capacity expansion. We now explicitly state this rationale.
>
> Role of pruning:
>
> We clarified that pruning is not proposed as a mandatory component of EB, but as a mechanism for analyzing expert allocation and specialization. Organ-wise pruning statistics at the deepest encoder stage are now provided in the Appendix and summarized in Sec. 4.3, directly addressing the concern about unclear motivation.
>
> Training protocol transparency:
>
> We added full optimization details (Adam optimizer, learning rate, weight decay, epochs, no early stopping) and clarified that all models are trained from scratch under identical conditions for fair comparison.

---

### Official Review · Reviewer_VdMi · 2026-01-09

**Confidence:** 4
**Preliminary Rating:** 4

**Summary:**

This paper introduces Expert Branches, a modular framework for laparoscopic segmentation that uses separate feature-extraction branches instead of a single mixed stream. Each branch specializes in different geometric patterns, improving feature diversity with minimal extra parameters. Implemented as TriEB on a UNet backbone and tested on the DSAD dataset without pretraining, the method outperforms UNet, single-operator, and transformer baselines in both region and boundary metrics. Overall, it provides a scalable approach for accurate surgical segmentation under limited data.

**Strengths:**

The paper presents Expert Branches, a clear and generalizable architectural idea that encourages module diversity through disentangled pathways, addressing limitations of entangled multi-branch designs. Its rigorously controlled experiments (no pretraining, fixed parameters) convincingly isolate the benefit of diversity, with fair comparisons to scratch-trained transformers. The evaluation is thorough and clinically relevant, including boundary metrics, organ-wise analysis, and pruning for interpretability. The work is well-written, well-situated in prior literature, and supported by strong ablations.

**Weaknesses:**

Validated only on TriEB (UNet), three convolutional experts, and one dataset (DSAD); applicability to other architectures, attention-based experts, or modalities (CT/MRI) is untested.
 Comparison with existing multi-branch and late-fusion designs is limited, leaving the novelty and advantages of interpretability and pruning under-argued.
 Pruning is proposed as “expert allocation,” but it lacks detailed analysis of which branches are pruned and why, weakening the interpretability claim.
 CNN, DCN, and DSC selection is reasonable but not shown to be optimal; there is no study of alternative or incremental expert compositions.
 Parameter savings are shown, but inference time, memory use, and training cost are not analyzed.
 The work is a strong proof-of-concept but falls short of fully validating EB as a mature, general framework.

**Detailed Comments:**

Ensure consistent citation formatting in the reference list. Some entries use "et al." (e.g., Dosovitskiy et al., 2021b) while others list all authors (e.g., Carstens et al., 2023). Adhere to the conference style guide.
The abstract uses first-person plural ("we introduce"), while the main text is largely third-person. Consider standardizing voice for consistency, typically third-person for scientific papers.

The description of the weighted fusion (Eq. 1, 2) is clear, but briefly defining GAP (Global Average Pooling) in the caption or text would aid readers not immediately familiar with the acronym.
Section 3.3, Pruning: The choice of a 30% pruning ratio appears arbitrary. A single sentence justifying this choice (e.g., "A 30% ratio was selected as a commonly used aggressive yet stable sparsity target in network pruning literature") would strengthen the methodology.
Table 1 Footnote: Specify that parameter counts are in millions. While implied, explicitly stating "(in millions)" in the table header or caption removes ambiguity.

Figure 2 Reference: The caption "Qualitative comparison of the methods.The images are selected to highlight improvements." is too vague. A more informative caption should briefly state what each column shows (e.g., Input, Ground Truth, UNet, TriEB-CC, etc.) and what specific improvement is being highlighted (e.g., "better boundary continuity for the colon").
Discussion of Transformer Baselines: When noting transformers' weaker performance from scratch, it could be briefly qualified that this is an expected result given their known data hunger, which reinforces the paper's motivation rather than being a critique of transformers per se.
Organ-Wise Analysis (Table 3): The text notes challenging performance on the ureter. Adding a brief speculative reason (e.g., "likely due to its extremely thin and frequently occluded structure") would enrich the discussion. Also, explicitly acknowledging cases where TriEB underperforms UNet (e.g., Intestinal veins for TriEB-WF) demonstrates balanced analysis.

Page 4, End of Paragraph 1: "...parameter growth associated with uniformly widening convolutional layers." The word "parameter" is split across lines ("parame / ter growth"). Ensure proper typesetting in the final version.
Page 7, Metrics Section: The sentence "In laparoscopic surgery, organ boundaries guide critical tasks..." begins a new line mid-sentence. This should be formatted as a continuous paragraph.
Throughout the text, perform a final proofread for minor grammatical issues (e.g., "The independent branches further exhibit interpretable specialization behavior," could be smoother as "The independent branches also exhibit...").

Figure 1: While described, ensuring the diagram in Figure 1 maximally contrasts "entangled" vs. "disentangled" pathways with clear visual cues (e.g., distinct colors for different module families) would improve its pedagogical value.
Future Work: The conclusion mentions "pretrained expert modules" and "more advanced surgical tasks." Being slightly more specific (e.g., "integrating pretrained vision transformer branches" or "extending to surgical phase recognition or tool segmentation") would give a clearer vision for follow-up research.

**Justification Of The Preliminary Rating:**

The paper merits acceptance because it successfully introduces and validates a novel architectural principle that addresses a clear gap in feature learning for data-scarce, geometrically complex domains like surgical segmentation. Its primary strength is the rigorous, controlled experimental design that isolates the effect of module diversity from confounding factors like pretraining and parameter scaling, a methodologically sound approach that provides clear, interpretable evidence for its core thesis. The work is timely, offering a framework for building more efficient, interpretable, and adaptable models at a time when community focus is shifting beyond pure scale. While the weaknesses identified (e.g., limited validation scope, superficial pruning analysis) are valid, they represent standard and constructive avenues for future work rather than fatal flaws; they do not undermine the paper's solid proof-of-concept contribution, clear narrative, and scholarly rigor. Therefore, the paper meets the bar for acceptance by presenting a well-motivated idea with compelling initial validation and significant potential to influence research in efficient network design for specialized vision tasks.

**Questions To Address In The Rebuttal:**

The EB principle is presented as architecture-agnostic, yet evidence comes solely from TriEB (UNet + 3 conv. experts). Please discuss:
The feasibility and planned experiments for integrating a non-convolutional expert into the EB framework without disrupting its core design.
How would you test the principle's effectiveness on another segmentation backbone (e.g., a Deeplab variant) or a different but related task (e.g., surgical tool segmentation) to substantiate the claim of generality?
 Pruning is framed as a tool for analyzing expert contributions. Please provide:
A preliminary analysis or discussion of which expert branches (CNN, DCN, DSC) were pruned more or less aggressively in different encoder stages or for different organ classes in your experiments. This data is crucial to support the interpretability claim.
The choice of CNN, DCN, and DSC is sensible but not explored comparatively.
Please comment on whether you conducted, or considered, an ablation that incrementally adds experts (e.g., CNN+DSC vs. CNN+DCN vs. all three) to demonstrate that the benefit is due to diversity itself rather than the specific trio chosen.
Discuss the potential role of automated neural architecture search (NAS) or a principled criterion for selecting complementary experts in future EB systems.

Please provide or estimate key operational metrics for TriEB-CC/WF versus the Vanilla UNet baseline, such as FLOPs, inference time (on a standard GPU), and training memory footprint. A discussion of how these trade-offs impact potential real-time surgical application is important for a complete assessment.
 Please elaborate on how the EB design, with its late and explicit fusion, differs from other multi-branch strategies that also aim to preserve module-specific features until later stages (beyond the cited ACmix). A more detailed comparison would help clarify the specific novelty in the flow of information your design enforces.

---

> ### Author Response · Authors · 2026-01-25
>
> We sincerely thank the reviewer for the positive evaluation and for the detailed, constructive feedback. Below we address the key points raised by the reviewer.
>
> Generality of Expert Branches (EB):
>
> We have strengthened the paper’s framing to clearly present EB as an architecture-agnostic design principle, rather than a UNet-specific construction. In the Introduction, Sec. 3.2, and the Conclusion, we explicitly state that EB can be applied to alternative backbones , and that the number and composition of expert branches are flexible. We further clarify that the current TriEB design is a deliberately constrained instantiation used to isolate the intrinsic effect of module diversity, not to define an optimal architecture. Concrete future directions now explicitly include integrating non-convolutional experts such as vision transformers, with each expert pretrained independently prior to fusion.
>
>
> Pruning and interpretability:
>
> We clarified that pruning is used as an analysis tool rather than a deployment requirement (Sec. 3.3). To directly support the interpretability claim, we added an organ-wise pruning analysis at the deepest encoder stage and report full statistics in Appendix A. The results show that CNN features are consistently preserved across organs (low variance), while DCN and DSC exhibit higher and organ-dependent sparsity. This demonstrates that expert branches contribute differently across anatomical structures and collaborate through complementary roles rather than redundancy.
>
> Efficiency and operational metrics:
>
> We added detailed training hyperparameters and inference-time discussion at the beginning of Sec. 4.1. All models share the same encoder–decoder depth and layer structure, resulting in millisecond-level inference times with negligible differences. This confirms that the observed gains arise from module diversity rather than increased computational cost.

---

> > ### Comment · Reviewer_VdMi · 2026-02-02
> >
> > I appreciate the authors for their clear and effective responses to the assigned comments.

---

### Author Rebuttal · Authors · 2026-01-25

**Rebuttal:**

The revised manuscript clarifies that Expert Branches (EB) are proposed as a general architectural principle rather than a specific network, and that the goal is not to pursue state-of-the-art performance. The study is explicitly framed as a controlled validation of how module diversity affects intrinsic feature learning under constrained conditions, with TriEB serving as a minimal and deliberately restricted instantiation. The main revisions are summarized below.

Backbone Choice and Generality.
The use of UNet is now explicitly justified as a methodological control that enables strict parameter scaling, ensuring that performance differences arise from module diversity rather than increased capacity. The manuscript further emphasizes that EB is backbone-agnostic, flexible in the number and composition of expert branches, and applicable to alternative architectures. Future extensions incorporating pretrained vision transformer experts are explicitly discussed.

Pruning and Interpretability.
Pruning is clarified as an analysis tool rather than a required component of EB. A new organ-wise pruning analysis at the deepest encoder stage is added, with detailed statistics provided in the Appendix. The results show that standard convolutional features are consistently preserved across organs, while deformable and dynamic snake convolution branches exhibit higher and organ-dependent sparsity, supporting complementary expert collaboration.

Statistical and Experimental Clarity.
Statistical significance indicators are added to the main tables, with significance evaluated across all 11 organs. The experimental section now reports full training details and clarifies that all models, including transformer baselines, are trained from scratch under identical settings with comparable inference time.

**Supporting Material:**

/attachment/fbf88b0c63af61c773794815251f65e92c8ddbfc.pdf

---

### Meta-Review · Area_Chair_CedJ · 2026-02-09

**Recommendation:** Accept (Poster)
**Confidence:** 4

**Metareview:**

The manuscript received highly diverging reviews in the initial round, ranging from strong accept all the way to strong reject. After revision, the positive reviewers remain positive while the most negative reviewer updated their score to borderline, stating a few remaining reservations especially around the possibly quite artificial setting of the study that may lack any real-world practicality as well as somewhat unconvincing results. While I personally share some of these reservations, the reviewer concensus now is positive which is why my recommendation is for acceptance as a poster.

---

### Decision · Program_Chairs · 2026-02-13

Accept (Poster)